# The Influence of Social Media on the Perception of Autism Spectrum Disorders: Content Analysis of Public Discourse on YouTube Videos

**DOI:** 10.3390/ijerph20043246

**Published:** 2023-02-13

**Authors:** Schwab Bakombo, Paulette Ewalefo, Anne T. M. Konkle

**Affiliations:** 1Interdisciplinary School of Health Sciences, University of Ottawa, Ottawa, ON K1N 6N5, Canada; 2School of Psychology, University of Ottawa, Ottawa, ON K1N 6N5, Canada; 3Mind Research Institute, University of Ottawa Brain, Ottawa, ON K1Y 4E9, Canada

**Keywords:** autism, autism spectrum disorder, Asperger, ASD, social media, YouTube, advocacy, awareness

## Abstract

Background: Little is known about how social media shapes the public’s perception of autism spectrum disorder (ASD). We used a media content analysis approach to analyze the public’s perception of ASD. Methods: We conducted a YouTube search in 2019 using keywords related to ASD. The first 10 videos displayed after each search that met the eligibility criteria were selected for analysis. The final sample size of videos analyzed was 50. The top 10 comments from each respective video were selected for commentary analysis. A total of 500 comments were used for this study. Videos and comments were categorized based on sentiment, evident themes, and subthemes. In 2022, using the same key words, we conducted a subsequent YouTube search using the same criteria, except that the videos had to be 10 min or less, whereby nine videos were selected out of 70 for commentary analysis, and a total of 180 comments were used. Results: The dominant themes were “providing educational information on ASD characteristics” with the main subtheme being “no specific age or sex focus”. The most common category of comments was “anecdote”. The overwhelming sentiments of both the videos and comments were “mixed”. Individuals with ASD were stigmatized as not being able to understand emotion. Furthermore, ASD was also stigmatized as being a monolithic condition only manifesting itself in the most severe form when autism varies in severity. Interpretation: YouTube is a powerful tool that allows people and organizations to raise awareness about ASD by providing a more dynamic view on autism and fostering an environment for public empathy and support.

## 1. Introduction

### 1.1. What Is Autism Spectrum Disorder?

Autism spectrum disorder (ASD) is a neurodevelopmental condition often distinguished by social communication deficits and repetitive sensory motor behaviours usually appearing early in life. ASD can also be characterized by maladaptive cognitive functioning and particular behaviours that impair communication and learning, thus affecting social interaction [1,2,3]. ASD is typically present before the age of 3 years and is commonly accompanied by abnormalities in cognitive functioning, sensory processing, attention, and learning [4]. ASD is considered a spectrum because the various cognitive, social, and behavioural manifestations differ greatly amongst individuals, and the degree of impairment may range from mild to severe [4]. Though the etiology of ASD is still obscure, the consensus is that the development of atypical neurological functioning is multifactorial, involving the complex interaction of genetic and environmental components [5]. 

ASD is quite common. It is estimated that 1/100 children are diagnosed with autism around the world [6]. In addition, ASD is up to four times more common in boys than girls [7] and can affect people of all races, ethnicities, and socioeconomic backgrounds [8]. Diagnosing ASD can be complex, as there is no specific test for this condition. With the use of the DSM-V, experienced healthcare professionals, such as pediatricians and psychiatrists, can make a clinical diagnosis based on behavioural observation [7,9]. There is no cure for ASD; however, it can be managed through early social and communication-based intervention programs, as well as speech therapy [3]. Medication is usually not administered for ASD but rather for accompanying diagnoses such as ADHD and anxiety [10].

### 1.2. Stigmatization towards ASD

In 1943, before legitimate research had been conducted, an Austrian American child psychiatrist, Leo Kanner, publicized that ASD was expressed due to the lack of “affective contact”. Based on this view, books such as The Empty Fortress (1967), by psychoanalyst Bruno Bettelheim, cited the emotional chilliness of “refrigerator mothers” to be the cause of ASD [11]. This theory was the first of many stigmatizing misconceptions towards individuals with ASD and their families.

The social misunderstanding of ASD can perpetuate stigma. Some of the public’s misperceptions include ideas such as limited work opportunities and diminished life skills [12]. Such stigmatizing attitudes can lead to discrimination, rejection, and exclusion from participation in many areas of society [13]. Furthermore, stigma negatively impacts an individual’s psychosocial well-being and can pose a barrier to seeking, accessing, and adhering to intervention programs [13]. Breaking these negative public perceptions requires the provision of accurate information, directive guidance on how to interact with peers with ASD, and endorsement of positive language, such as “people with ASD can go to college and get married” [14]. On that basis, the media can be underscored as an area where these significant adjustments can be made.

### 1.3. The Media’s Influence on Public Perception

The media is the general public’s most important source of information about mental health conditions and cognitive neurodivergences, such as ASD [15]. It not only serves to mirror public opinions but also to take part in molding them [16]. The media’s negative portrayal of psychiatric conditions may lead to the reinforcement of stigma and stereotypes; however, the media may also contribute to increasing the recognition and understanding of these conditions which may improve public attitudes [17]. 

Social media should be highlighted as an area where one can discern society’s current perspectives on ASD. These online communication platforms allow community-based input, content sharing, interaction, and collaboration [18]. Information, whether facts or opinions, circulated on social media platforms, such as Facebook, Reddit, Twitter, TikTok, and YouTube, have the power to influence public perception regarding ASD. While there exist studies that have assessed the representation of individuals’ experience with ASD on YouTube, to the best of our knowledge, this study is unique in its analysis of content together with the public’s comments on ASD.

### 1.4. YouTube as a Platform for ASD Awareness

In recent times, awareness about mental health and neurodiversity has been on the rise and social media has become a megaplatform for spreading this knowledge. YouTube, a video-sharing platform, is a popular place for these discussions. With their motto of “Broadcast Yourself”, YouTube has over 1.9 billion users visiting their site each month to view hours of video content (YouTube for Press, n.d.). YouTube videos have become a powerful tool for raising awareness and offering alternate ways of thinking about ASD [19,20]. 

Notwithstanding the fact that the average number of views for the videos retrieved in 2019 was 826,156 and those retrieved in 2022 was 383,111, with some videos reaching millions of views, YouTube remains a go-to resource for viewing a variety of video types. Since its inception in 2005, it is one of the most visited sites in the world. Its content is diverse and global, and the platform offers the opportunity to disseminate content to a very broad audience [21]. YouTube is an attractive platform for both amateur content creators and media companies, politicians, news organizations, businesses, music and film artists, and especially educational institutes [21], with many channels related to health—a quick search for ASD and YouTube shows over 33 million video results. 

Stories shared by individuals with ASD or their parents, siblings, and friends provide an assortment of insight and personal experiences. Furthermore, YouTube has become a place for many organizations and healthcare professionals to upload factual information about ASD. These videos and their commentary sections are of immense value for individuals who are looking to increase their knowledge and understanding of ASD or find people from across the globe who share like experiences [22]. The practical components of YouTube, such as the availability of free content 24/7 and the greater appeal of videos compared to articles or pamphlets, make this platform an exceptional way to interact with an audience [19,23]. 

### 1.5. Aim

Given that positive and appropriate media representations are key to improving public perceptions about mental health conditions and neurodiversity [24,25,26,27], the objective of this work was to analyze the portrayal and perception regarding ASD by YouTube content uploaders and commentators. 

There is a gap in the literature when it comes to the analysis of content together with the public’s reaction to ASD. More evidence is needed to better inform how content with comments analysis on YouTube videos can improve the public’s perception of ASD. This will be achieved by answering the following research questions:

(1) What are the prevalent themes and sentiments in ASD related videos and their comments? (2) Do video presenters and commentators display stigmatization towards ASD?

## 2. Methodology and Framework

We conducted a conventional content analysis to identify the themes and tonality used in videos and their related comments on ASD appearing on YouTube. The methodological framework used was that of phenomenology, which seeks to describe and interpret theory of social action and to analyze subjective experience and social relationships within the daily life of individuals [28]. The subjective meaning of people’s experience drove the research conducted through YouTube [29]. We collected YouTube videos and analyzed them prior to developing themes. Analyzing the subjective perspectives of the public was the main goal for collecting data [30]. A deductive approach was used in order to drive the research questions; phenomenology can contain the process of deductive thematic analysis, hence having themes emerge from collected data via inductive coding [29]. We did not use a codebook in this analysis, even though videos with more than one attribute were duly coded.

We stated the hypothesis that social media such as YouTube does have an influence on the public’s perception of autism. We also went on to see whether the pandemic changed the public’s perception of ASD. YouTube (www.youtube.com (accessed on 30 April 2019)) was searched for ASD-related videos which were categorized by basic descriptors and examined for content. For the purpose of this study, the search was conducted without signing into a personal account in order to reduce the influence of previous searches on the resulting videos. Due to the public nature of YouTube posts and anonymity in reporting our findings, no approval for ethics was required. 

### 2.1. Key Search Words

The following terms were inputted in the YouTube search bar: autism, autism spectrum disorder, autistic disorder, autistic behaviour, and Asperger’s. These terms are commonly used words when ASD is referenced in numerous articles and videos. The “ASD” abbreviation for autism spectrum disorder was omitted from the key search words because ASD is also short-term for atrial septal defect and a name for a musical group. Therefore, a disproportionately low number of videos related to autism were displayed.

### 2.2. Inclusion Criteria

Videos were included in this study if they were related to the topic of ASD. In addition, videos had to be in the English language and have clear and audible sound quality. Furthermore, the comment section had to be enabled and videos had to have at least 100 comments. There seems to be a positive relationship between popularity and comments in terms of user–content interaction [31]. Thus, this criterion serves as a buffer for those viewers selecting videos to watch based on popularity. Lastly, the duration of the video had to be 20 min or less for those retrieved in 2019 and 10 min or less for videos retrieved in 2022. The duration was restricted because we anticipated that viewers are less likely to watch extremely long videos. Only comments in the English language were considered for analysis. 

### 2.3. Selection Procedure

Since YouTube does not display the number of search results, we opted to analyze the first 10 videos displayed with each key term. If a video did not meet the eligibility criteria or if there was a duplicate, then the next video in the list of results was selected. 

YouTube was searched, in 2019, using the advanced settings to display the most relevant videos in descending order. The preference to analyze the most relevant videos instead of the most viewed emanated from the fact that the YouTube filter settings were automatically set to “sort by: relevance”. People usually do not play with the filter settings before looking for videos; thus, the content displayed to them first will be the “most relevant” and not necessarily the “most viewed”. A subsequent YouTube search was performed in 2022, using the same set of criteria and sorting preferences, from which 9 videos, lasting 10 min or less, were included for content and thematic analysis only as far as themes and subthemes in the top 20 comments of each video. This was conducted to assess whether the pandemic may have shifted public perception, hence warranting a further study on the same topic for videos released during COVID-19.

### 2.4. Basic Descriptors and Attributes

The following information was collected from the videos: date of upload, length of video, number of views, number of comments, as well as the labelling categories used by YouTube for each video. The following attributes were also noted for the 2019 videos: apparent sex of the individual with ASD in the video (male or female), apparent race of the individual with ASD in the video (white, black, Asian, or other), age category of the individual with ASD in the video (child or adult), and finally the presence of a healthcare professional, such as a physician, nurse, or therapist (yes or no). Videos with more than one attribute were duly coded.

### 2.5. Identification of Themes

The videos and comments were categorized based on their overarching themes. The methodology of Madden, Ruthven, and McMenemy [32] was adapted to define and categorize themes in the comments section. If a comment or video addressed multiple themes, it was classified by its most dominant theme. Quotes were selected to illustrate each theme in the comment section. 

### 2.6. Identification of Sentiment

The sentiment of the videos and comments was also recorded. Since YouTube videos provide visual and spoken messages, the sentiment was assessed by the expressions of the individuals in the video and the impression it left with the viewer regarding ASD. A video was categorized as positive if the individuals in the video were energetic and cheerful and the video left the viewer with a more positive impression of ASD. A video was classified as negative if the individuals in the video appear exhausted, angry, or frustrated and the video left a more negative impression of ASD on the viewer. Mixed sentiments were identified as those that highlight both positive and negative aspects and leave a reader with a more dynamic view of ASD. Comments were also categorized as positive, negative, or neutral in sentiment using the Azure machine learning add-in for Excel. Microsoft Azure machine learning was used to perform sentiment analysis and to develop a classification model that allowed for the identification of sentiments [33]. It is now known that sentiment analysis is a widely used technique in the natural language processing realm for determining the sentiment of a text, especially in social media [33].

### 2.7. Identification of Damaging Language and Stigmatization

We also evaluated the videos for the use of damaging language and stigmatization. With respect to damaging language, we assessed whether individuals were referring to people with autism as “suffering” and not using person-first language. There is no public consensus concerning the way autism is and should be described, especially in that people use many terms to describe autism [34]. An example of person-first language would be “children with autism” instead of “autistic children”. Moreover, the Autism Speaks organization website provides a plethora of stigmatizing remarks held towards ASD. The examples presented below were used to identify stigmatization present in YouTube videos and comments (“11 Myths about Autism”, 2018): People with autism do not want friends;People with autism cannot feel or express any emotion—happy or sad;People with autism cannot understand the emotions of others;People with autism are intellectually disabled;People who display qualities that may be typical of a person with autism are just odd and will grow out of it;Autism only affects children;Autism is just a brain disorder.

## 3. Results

### 3.1. Basic Descriptors for 2019 and 2022 Videos

In 2019, we identified 50 videos that met the criteria for eligibility and extracted 500 comments from a sample of 99,316 for analysis. The date of upload for the videos ranged from 29 September 2010 to 29 November 2018. The videos had a mean number of views of 826,156 (range: 9010–8,651,424), were on average 8:53 (min:sec) in duration (range: 1:34–20:00), and had an average of 1986 (range: 126–11,309) comments per video. 

In 2022, we identified nine videos that met the criteria for eligibility and extracted 180 comments from a sample of 19,062 for analysis. The date of upload for the videos ranged from 13 March 2020 to 30 October 2022. The videos had a mean number of views of 383,111 (range: 10,000–1,400,000), were on average 6:70 (min:sec) in duration (range: 1:59–10:37), and had an average of 2710 (range: 48–5825) comments per video.

### 3.2. Attributes for Videos Selected 

To assess the attributes of age, race, and sex, we analyzed videos that had an individual with ASD present in the video. This excluded animated videos of people with ASD and videos that refer to, for example a male child with ASD, but do not include pictures or short clips of the child. Out of the 59 videos gathered, 36 videos (61%) had an individual with ASD present. Table 1 shows that children and adults were equally present in the videos, with males being more prominent and apparent White as the race most often seen. Table 2 displays the percentage of videos in the labelling categories used by YouTube, with People and Blogs being the most represented category. We also discerned that 78% of the videos did not have a healthcare professional present. 

## 4. Themes in 2019 Videos

This content analysis found that the discussion of ASD-related topics was centralized around three themes: providing educational information, discussing personal experiences, and displaying daily life. The dominant theme within the videos was providing educational information on ASD, with presenters having no specific age or sex focus. Out of the 28 videos that were dedicated to providing educational information, only 11 videos (39%) included a healthcare professional, such as a psychiatrist, nurse practitioner, speech pathologist, or occupational therapist. Figure 1 shows the distribution of themes and subthemes within the 50 videos analyzed. 

Providing Educational Information (57.2%): The content in videos was constructed to provide educational information on ASD. This included a discussion of ASD etiology, social and behavioural manifestations, diagnosis, and interventions. This category can be further classified by the demographic of the people with ASD the video focused on:
1.1.Specific focus on children (18.4%): Videos discussing social and behavioural manifestations a child might exhibit, for example, self-stimulatory behaviours, such as hand flapping, grunting, or excessive blinking. Videos with this theme also included animations to help children learn about ASD.1.2.Specific focus on adults (4.1%): Videos discussing topics such as communication difficulties and impairments in abstract thought that might persist into adulthood, in addition to addressing how to attain a late diagnosis.1.3.Specific focus on females (2.1%): Videos underscore ASD characteristics in females, issues with the lack of female ASD research, and how women are being misdiagnosed with other conditions such as bipolar disorder.1.4.No specific age or sex focus (32.6%): Content in these videos is generic. In some videos, the presenter does not direct ASD-related information to a specific demographic. In others, the presenter may highlight attributes of ASD in males, females, children, and adults.Discussion of Personal Experiences (30.5%): Videos with this theme have individuals with ASD or their significant others (child, parent, coworker, friend, etc.) share their own personal experience of having ASD or taking care of or interacting with a person with ASD. Three subthemes emanated from this category:
2.1.Personal Social and Behavioural Manifestations (12.0%): Video presenters provide background information on the social and behavioural manifestations that they or their significant other exhibit. They also elaborate on how these neurological differences have affected their lives; for example, some individuals note that it made social interaction more difficult. Further discussions included their age of diagnosis and interventions they have used.2.2.Public Perception (9.8%): Video presenters gave an account on the things people have said to them or how their friends, family, and peers have treated them or a significant other with ASD. For example, people say “you do not look autistic”. Some presenters also specifically highlight incidences of bullying from their peers, as well as others misunderstanding their behaviour and thinking they are “rude” or “weird”.2.3.Self Concept (8.7%): Individuals in the video discuss how they felt about having ASD before and after a diagnosis. Individuals discussed feeling awkward or like they were not normal. Some expressed feeling lonely and self-esteem issues, such as not feeling like they were beautiful. In addition, it was mentioned how an actual diagnosis brought relief and comfort to their life.Daily Life Videos (12.3%): Videos with this theme included vlogs of people showcasing activities performed during their day. This category excludes videos that have presenters displaying and explaining behaviours in order to educate their audience. This theme is broken down into two subthemes:
3.1.Completing daily routines (4.1%): These videos display individuals completing day-to-day activities, such as getting ready for school, travelling, and attending doctors’ appointments.3.2.Meltdowns (8.2%): These videos specifically highlight an incident during the day. They display an individual with ASD having a “meltdown”. These videos showcase self-injurious behaviour and uncontrolled screaming or crying.

## 5. Themes in Comments

Using an adapted modified methods classification scheme for content analyses of YouTube video comments [32], we found that the dominant theme of YouTube video comments was “anecdote” at 39.8%. Many commentators provided an example of their own personal experience of having ASD or that of a significant other. A majority highlighted the social and behavioural manifestations that they exhibit. Figure 2 displays the stratification of the themes and subthemes in the 500 YouTube comments analyzed. A similar figure was not needed for the 2022 videos, since only nine were analyzed compared with 50 for the 2019 retrieved videos. Table 3 provides a breakdown of comment themes and subthemes, as well as illustrative quotes, for the videos retrieved in 2019, while Table 4 provides a breakdown of comment themes and subthemes with quotes for videos retrieved in 2022.

## 6. Sentiment of Videos and Comments

The majority of videos (70%) had mixed sentiments. The video presenters highlighted both positive and negative aspects of ASD and tried to provide their audience with a balanced perspective. However, comments on the videos were mostly negative in sentiment (59.4%); this trend remained for the comments on videos collected in 2022 with most comments being negative in sentiment (76.9%). 

## 7. Use of Damaging Language and Stigmatization for 2019 Retrievals 

The use of damaging language was minimal, with 8% of videos and 3.5% of comments not using person-first language. No video presenters or commentators referred to individuals with ASD as “suffering”. Likewise, the prevalence of stigma was very low. Only 2% of videos and 0.6% of comments were flagged as stigmatizing. A stigmatizing remark made by video content creators and commentators was that persons with ASD cannot experience emotions or that they lack empathy. Furthermore, ASD was also stigmatized as being a homogenous condition, with only the most severe form of ASD being characterized as “true autism”.

## 8. Discussion

There is a multitude of research that assesses the representation of ASD in film, literature, and newsprint; “However, analysis of social media sources has not been employed widely, if at all, in the context of ASD” (Lloyd et al., 2019). Social media has the power to bring awareness to ASD or skew public perception. Thus, the goal of this study was to examine how ASD was depicted in YouTube videos and their associated comments. 

### 8.1. Video Attributes: Major Findings and Implications

Most of the videos retrieved in 2019 (62.8%) showcased a male individual with ASD. This corresponds with epidemiological prevalence rates because there is a higher ratio of males to females with autism [35]. There was also an almost equal prevalence of adults (51.2%) and children (48.8%) who disclosed having ASD. This balanced distribution is important to show because there is still public perception that ASD only occurs in childhood and is not a lifelong condition (“11 Myths about Autism”, 2018). 

With respect to the demographic data pertaining to ASD, the race of the individuals shown to have ASD was almost all White. The implications of having most videos showcase ASD in Caucasian individuals is that it makes it appear as though ASD is unique to Caucasians. Scientifically, this is not the case, as the ASD phenotype is considered to not differ by race [36]. This low representation of ASD in minority populations may reflect the under diagnosis or misdiagnosis of individuals within these racial groups, while the existing gaps in health equity and accessibility remain. For example, ASD identification in black children occurs at older ages, and they are more likely to be diagnosed with an adjustment or conduct disorder [37] rather than ASD. Delayed diagnoses can hinder early intervention toward better health outcomes for people with ASD [38]. Moreover, it is important to have diverse racial representation so that the perception of who can have ASD is not skewed. Finally, hearing and viewing individuals discuss their ASD-related diagnoses could encourage others of the same racial background to potentially seek medical assistance or support if needed. 

### 8.2. YouTube Videos as an Educational Resource

The most prominent theme that we found in all videos concerned providing educational information. It is very beneficial to have an assortment of authors publicize information pertaining to characteristics of ASD. A study by Fusaro et al. [39] demonstrates the feasibility of using YouTube videos for accelerating the early detection of ASD. Their results showed that ASD-related behaviours identified in nonclinical videos could be used to correctly detect the presence of autism [39]. Despite nonclinical videos having factual information, the presence of healthcare professionals in these videos is still needed. Only 22% of the sample 2019 videos were presented by a healthcare professional. Their inclusion in ASD-related discussions is important because the public needs evidence-based material from perceived reputable sources. Physicians, psychologists, educators, nurses, and occupational therapists all have the obligation to provide current, valid, and high-quality data to the public to ensure early ASD detection and intervention [40]. 

### 8.3. Benefits of Sharing Personal Stories in Videos and Comments

Anecdotal/personal experience was the principal theme in the comment section and the second most prominent of the 2019 and 2022 videos. The content analysis for the 2022 videos also found that the discussion of ASD-related topics focused on the same three themes, being providing educational information, discussing personal experiences, and displaying daily life. The videos’ overarching theme also concerned providing educational information on ASD, and there was no specific age or sex focus.

Social media provides an open space for people to discuss aspects of their lives such as having or parenting a child with ASD [41]. In the online community, individuals can give and seek advice and support. The results of a 2014 study found that the communal nature of social media enabled people to share coping strategies, find hope, and feel less isolated [42]. These elements were all present in the YouTube videos and comments. Under this theme, the hope of many individuals was that sharing their experiences would help or comfort others in similar situations. Since the neurological differences in ASD vary, accounts from people on different ends of the spectrum are critical to the improvement of public perception. 

### 8.4. “Mixed” Messages Provide a Dynamic View 

Contrary to our expectation, most videos had a “mixed” sentiment instead of a positive sentiment. This is understandable because many videos, those retrieved in 2019 and 2022 alike, were dedicated to increasing the general public’s knowledge and comprehension of ASD. Raising awareness does not necessarily have to include making ASD out to seem untroublesome, easy to overcome, or beneficial. Highlighting both positive and negative attributes can have more progressive effects on improving the public perception of ASD [43]. 

### 8.5. Severe Autism and Negative Comments

Six videos retrieved in 2019 (12%) were classified as negative. Two videos had the theme “discussing personal experience”, and the remaining four videos had the theme “daily life”. It is important to note that in all of these videos, the individual with ASD appears to be on the more severe end of the spectrum. They show behaviours such as uncontrolled shouting or crying, and they are also seen to be physically hurting themselves or others.

Since YouTube accounts can use pseudonyms, this helps provide a degree of anonymity. Thus, individuals may feel less restricted in expressing their genuine opinions or feelings towards ASD. They can make statements without fear of being identified especially when their comments are callous or bigoted. Though most videos were mixed in sentiment, negative videos do have the power to elicit negative responses for both pre- and pandemic videos. However, comments were highly negative in sentiment. Commentators were expressing negative sentiments towards behaviours displayed by individuals on the severe end of the spectrum. They were also dismayed by the caretaker’s (a parent or sibling) inability to manage that individual’s behaviour. Though content creators want to illuminate negative aspects of ASD, it is integral to inform the audience regarding plausible methods, treatments, and therapies to overcome some of the behavioural manifestations and perhaps symptoms associated with comorbid disorders. 

### 8.6. Stigmatization and Person-First Language

Stigmas, such as the perception that people with ASD cannot express or understand emotion, can impede the development and integrity of social relationships. Moreover, stigmas such as “severe ASD is the only form of ASD” can delay the early detection and diagnosis in those with milder social and behavioural manifestations. Therefore, content creators should address and resolve any misconceptions that operate to paint these individuals in a dark light or obstruct them from seeking medical attention, while seeking to increase public knowledge mobilization to improve the social outcome for those affected. 

There were few instances in videos and comments where people did not refer to individuals with ASD appropriately in both 2019 and 2022 videos. It is imperative to put a person before their condition using appropriate and sensitive terminology. “Person-first” language emphasizes the person rather than the impairment and helps preserve their humanity while promoting their individuality. For example, the Canadian Association of Broadcasters [44] encourages the use of person-first language, as they recommend broadcasters to state that a “person has a disability” rather than a “person is disabled” [44]. Doing this will affirm their identity as a human first, whilst dissociating the stigmas associated with the condition from the person.

### 8.7. Limitations

Since this study only focused on YouTube, not reviewing videos on other social media and video-sharing sites is a limitation of this study. In addition, the IP address used in the collection of videos could have been a factor in directing our search results to mainly North American videos due to the failure of adjusting for geo-filters. Furthermore, because we did not contact the individuals in the videos, we cannot confirm sex, age, or race attributes. There were no duplicate videos and ASD-related topics, and the perspectives were unique and varied between each video. However, multiple entries by the same YouTuber (7 out 50 videos retrieved in 2019) is a limitation to the sample size of the study. Comments from videos retrieved in 2022 were reviewed to see whether the pandemic may have shifted public’s perception on ASD. The video content analysis was limited for 2022 videos. By not logging in, we are mindful that this may have limited our search results, even though we believed that signing into a personal account may increase the influence of previous searches on the resulting videos. Lastly, for all videos, the comment section is subject to bias. Content uploaders can delete comments that they think are harassing or abusive. The possibility of deleted negative responses was a limitation as to the analysis of sentiment in the comment section. 

### 8.8. Future Studies

The results of this study lend themselves well to expansion by integrating and potentially comparing videos from multiple countries on various video sharing sites, by systematically reviewing pre- and post-pandemic videos, and conducted a step-by-step comparison. The use of statistical analysis software may provide an additional layer in providing a summative account of video sentiments or themes to the types of comments. The results reported herein suggest that healthcare professionals, as well as minority racial and ethnic groups, should aim to have a stronger online presence while considering that culturally targeted health information and interventions are greatly needed to mend the gap in health equity and accessibility. In addition, individual content creators should be well versed in the use of person-first language. Advocacy organizations can create guidelines to be used on YouTube and other social media platforms to properly address individuals with mental health conditions.

## 9. Conclusions

In conclusion, this study identified the utilization of YouTube as a tool for education and a platform for people to share stories about their ASD-related experiences and interact with others. We also determined that the information and perspectives shared in videos retrieved in 2019 and 2022 provide a more dynamic view of ASD. Based on the results of the analysis, it also appears that the pandemic did not significantly shift public perception of ASD. More research, including systematic reviews, is critical to understanding how social media shape the public’s perception of ASD while offering to improve content and information sharing. Expanded clinical research, including culturally targeted interventions, and modified therapies, which may help improve social perceptions of ASD, is critical to improving outcomes for individuals with autism and their families.

## Figures and Tables

**Figure 1 ijerph-20-03246-f001:**
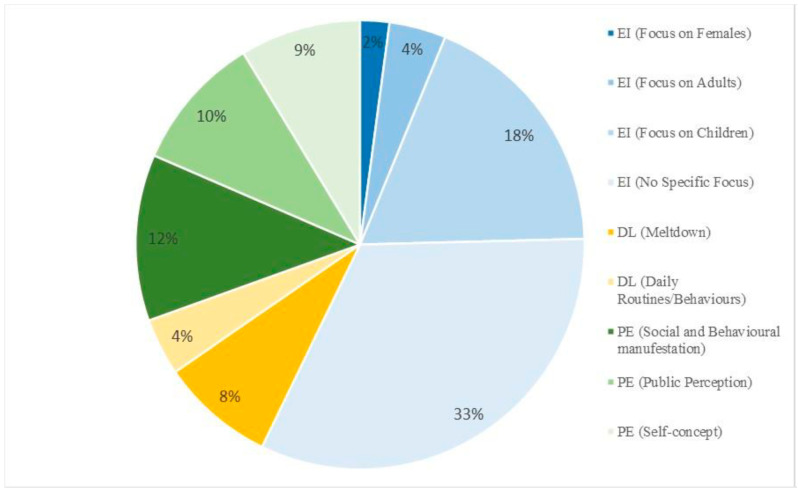
Major themes and subthemes presented in the ASD-related videos. N = 50. EI represents the theme “providing educational Information”. PE represents the theme “personal experience”. DL represents the theme “daily life videos”. In brackets are the subthemes for each.

**Figure 2 ijerph-20-03246-f002:**
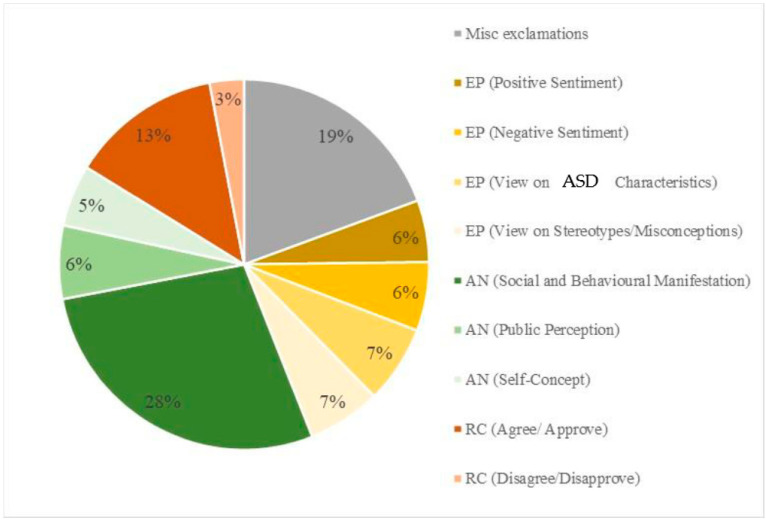
Major themes and subthemes presented in comments related to the chosen videos. Sample size = 500. EP represents the theme “expression of personal feelings”. AN represents the theme “anecdote”. RC represents the theme “reaction to video content”.

**Table 1 ijerph-20-03246-t001:** Percentage of individuals with ASD, with respect to age, sex, and race. n = 31.

**Age Category of Individual with ASD**	**(%)**
Child	48.8
Adult	51.2
**Sex of Individual with ASD**	**(%)**
Male	62.8
Female	37.2
**Race of Individual with ASD**	**(%)**
White	86.1
Black	9.3
Asian	2.3
Other	2.3

**Table 2 ijerph-20-03246-t002:** Distribution of videos in 10 different categories as labelled by YouTube. n = 59.

Percentage of Videos in Different YouTube Categories	(%)
Nonprofits and Activism	10
Education	26
People and Blogs	42
Science and Technology	4
Film and Animation	2
Gaming	2
Entertainment	6
News and Politics	4
Comedy	2
How to and Style	2

**Table 3 ijerph-20-03246-t003:** Themes and subthemes in YouTube comments with example quotes.

Theme or Subtheme	Description	Example Quotation
Anecdote	A comment containing a story about a commenter’s personal experience.	
Social and Behavioural Manifestations	A comment that shares the ASD-related social and behavioural manifestations of the writer or the writer’s significant other.	“I have Asperger’s and although I am a little socially awkward I’m quite extroverted which makes it hard to spot”.
Public Perception	A comment that provides an account of how the writer or their significant other might have been viewed or treated by others.	“People bully me at school. They say I’m weird because I’ll randomly zone out during class. I try to talk to them but they just say look at me then I’ll talk to you, and when I do look at them they tell me to stop staring at them”.
Self-Concept	A comment that provides an account of the writer’s feelings about themselves, including their self-image and issues with self-esteem.	“Having Asperger’s sucks. I feel like I’m a different species from the people around me”.
Expression of Personal Feelings	A comment in which the writer expresses their personal feelings, emotional response, or opinions/perspectives.	
Positive Sentiments	A comment that expresses the writer’s positive beliefs and attitudes towards ASD.	“People I have met that have autism are some of the most kind hearted empathetic and genuine humans”.
Negative Sentiments	A comment that expresses the writer’s negative beliefs and attitudes towards ASD.	“I would not want to have a child with autism”.
View on Stereotypes or Misconceptions	Comments that express a writer’s view on stereotypes or misconceptions held by society and perpetuated by the media or other information channels.	“I hate how the only autism portrayed in the media is the aesthetic kind where they turn out to be geniuses no one knows about severe autism it kills me”.
View on ASD Characteristics	Comments that express a writer’s view on the epidemiology of ASD, associated behaviours, interventions, etc.	“I don’t think it’s more common in boys. Because girls tend to hide it more often which probably means there are loads of girls out there with autism including me”.
Reaction to Video Content	Comments expressing a writer’s immediate reaction and feelings toward the information in a video or the way the creator presents it.	
Agreeing with or Approving of Creator Content	A comment that expresses agreement, satisfaction, or gratitude towards the video content.	“This was the best way to show neurotypical people autism I’ve ever seen. This is wonderful. Thank you”.
Disagreeing with or Disapproving of Creator Content	A comment that expresses disagreement or condemnation of video content.	“I don’t like this one bit. Those children are not autistic they may just have different personalities”.
Miscellaneous Exclamations	Comments with weak or no correlation to ASD-related content. This category also includes summarizing or directly quoting something in the video, making video requests, and random off-topic jokes, statements, or questions.	“Well now I have a crush on the girl in the maroon shirt”.

**Table 4 ijerph-20-03246-t004:** Themes and subthemes in YouTube comments with example quotes (2020–2022).

Theme or Subtheme	Description	Example Quotation
Anecdote	A comment containing a story about a commenter’s personal experience.	“The main problem with being autistic in today’s society is that the world just isn’t built for us”. This destroys me every day.
Social and Behavioural Manifestations	A comment that shares the ASD-related social and behavioural manifestations of the writer or the writer’s significant other.	“The inner conflict is maddening. All the things that make me feel most alive quickly burn me out…”
Public Perception	A comment that gives an account of how the writer or their significant other might have been viewed or treated by others.	“I’ve been hurt so many times by people I thought would never hurt me. Now that I’m 37 and have been crushed over and over with another one coming soon, I would just assume never get involved with another person”.
Self-Concept	A comment that gives an account of the writer’s feelings about themselves, including their self-image and issues with self-esteem.	“Nope, I’ve learnt through bad experiences that having friends is too much hassle. They always seem to turn on me after a while. Having friends is appealing in principle, but it just doesn’t work out. I am very lonely, but that’s preferable”.
Expression of Personal Feelings	A comment in which the writer expresses their personal feelings, emotional response, or opinions/perspectives.	“As someone with autism, I can honestly say I don’t care much about having friends. That wasn’t always the case, as I’ve gotten older, I seem to care less about having friends, but I think that’s normal for a lot of people”.
Positive Sentiments	A comment that expresses the writer’s positive beliefs and attitudes towards ASD.	“My 14 year old niece has both. She was diagnosed with ASD last year, but was diagnosed with ADHD when she was much younger... and we’ve always been incredibly close”.
Negative Sentiments	A comment that expresses the writer’s negative beliefs and attitudes towards ASD.	“I have Asperger syndrome, but you wouldn’t really notice unless I told you or if you study my behavior closely…and love being social”
View on Stereotypes or Misconceptions	Comments that express a writer’s view on stereotypes or misconceptions held by society and perpetuated by the media or other information channels.	“I’m 54 and I have lifetime adhd and asperger’s and it was hell trying to function most of my life as I had never even heard of either one until my late 30’s... as a child back in the 70’s I was diagnosed as hyperactive but that only scratched the surface of my issues”.
View on ASD Characteristics	Comments that express a writer’s view on the epidemiology of ASD, associated behaviours, interventions, etc.	“I’ve heard people complaining about schooling and working from home, via internet, while I’m like “Are you kidding? This is heaven!”
Reaction to Video Content	Comments expressing a writer’s immediate reaction and feelings toward the information in a video or the way the creator presents it.	“As an autistic person I want to say this for all others out there. We are built different, not incorrectly”.
Agreeing with or Approving of Creator Content	A comment that expresses agreement, satisfaction, or gratitude towards the video content.	“Our son just got diagnosed this week which has been such an amazing moment for our family. Really thankful for your content. Had a bit of a cry listening to your Podcast afterwards. Tears of relief and hope. Cheers?”
Disagreeing with or Disapproving of Creator Content	A comment that expresses disagreement or condemnation of video content.	“I have a autistic little brother and he’s 7 he can’t speak but tries to communicate he also has repetitive behaviour but Ty for saying this vid means a lot”.
Miscellaneous Exclamations	Comments with weak or no correlation to ASD-related content. This category also includes summarizing or directly quoting something in the video, making video requests, and random off-topic jokes, statements, or questions.	“This guy created Task manager, what a legend. All love from programmers around the world! thanks dave”.

## Data Availability

Publicly available data were analyzed in this study.

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
