# Peer review of "The Influence of Social Media on the Perception of Autism Spectrum Disorders: Content Analysis of Public Discourse on YouTube Videos"

_ijerph, 2023, doi:10.3390/ijerph20043246_

Round 1

Reviewer 1 Report

It's a very interesting article. Some methodological issues should be clarified, since in results-discussion a software is mentioned, but it is not explicit in the methodological framework.
There are also a couple of errors in Table 4, in rows 5 and 11. Typographical errors.

Author Response

It's a very interesting article. Some methodological issues should be clarified, since in results-discussion a software is mentioned, but it is not explicit in the methodological framework.
There are also a couple of errors in Table 4, in rows 5 and 11. Typographical errors.

Answers to Review 1:

  • We would like to thank you for your kind comment. We have addressed the issues that were outlined:
    • The mention of the Azure Machine Learning add-in for Excel is at the end of our methodological section right before the results section, on page 10. We have added a few lines to explain why this software is essential for this purpose.
  • Typological errors (in red) were corrected in Table 4, in rows 5 and 11

Reviewer 2 Report

 The Influence of Social Media on the Perception of Autism Spectrum Disorders: Content Analysis of Public Discourse on Youtube Videos

Thank you for the opportunity to review this interesting paper. My recommendation is to accept with minor changes. Here are some comments that need to be addressed.

This paper appears to adopt a medical model perspective on ASD. The language is replete with terms such disorder, symptoms, and even the term patient is used on line 52, with the term ‘treatment’ occurring 4 times. 

To make the paper more acceptable to a wider audience, they should take into consideration other perspectives on ASD. These references will be helpful

Leadbitter, K., Buckle, K. L., Ellis, C., & Dekker, M. (2021a). Autistic Self-Advocacy and the Neurodiversity Movement: Implications for Autism Early Intervention Research and Practice. Frontiers in Psychology12. https://doi.org/10.3389/fpsyg.2021.635690

Leadbitter, K., Buckle, K. L., Ellis, C., & Dekker, M. (2021b). Autistic Self-Advocacy and the Neurodiversity Movement: Implications for Autism Early Intervention Research and Practice. Frontiers in Psychology12. https://doi.org/10.3389/fpsyg.2021.635690

Milton, D. (2012). The normalisation agenda and the psycho-emotional disablement of autistic people. In Autonomy, the Critical Journal of Interdisciplinary Autism Studies (Vol. 1, Issue 1).

Silberman, S. (2015). Neurotribes: The legacy of autism and how to think smarter about people who think differently. Allen & Unwin.

Lines 51-52

Regarding the term ‘patient’ on lines 51-52, it should be clarified that it is not the patient, but their behaviour that is used for making a diagnosis. This should be corrected.

111-112

It is implied here that ASD is a mental health condition, which it is not.

126

Tobin and Begley font needs resized.

202

With respect to damaging language, the work by Kenny et al (2016) should be referenced here. This is important for the paper and for the analysis of terms. There may be cultural differences that need to be addressed.

“Recent public discussions suggest that there is much disagreement about the way autism is and should be described. This study sought to elicit the views and preferences of UK autism community members – autistic people, parents and their broader support network – about the terms they use to describe autism. In all, 3470 UK residents responded to an online survey on their preferred ways of describing autism and their rationale for such preferences. The results clearly show that people use many terms to describe autism. The most highly endorsed terms were ‘autism’ and ‘on the autism spectrum’, and to a lesser extent, ‘autism spectrum disorder’, for which there was consensus across community groups. The groups disagreed, however, on the use of several terms. The term ‘autistic’ was endorsed by a large percentage of autistic adults, family members/friends and parents but by considerably fewer professionals; ‘person with autism’ was endorsed by almost half of professionals but by fewer autistic adults and parents.

Kenny et al. (2016). Which terms should be used to describe autism? Perspectives from the UK autism community. Autism, 20(4):442-62. DOI: 10.1177/1362361315588200

240

Text in left-hand column is difficult to read when centered.

243-246

Use of capitalisation here is very odd.

Figures 1 & 2

The font is the pie chart could be reduced in size and this would allow the Key abbreviations to added to the chart. Not all Key terms are included in the figure caption.

256-257

“This 256 category can be further classified by the target audience of the video creator.”

Not clear what this means. Also, check the tense used in the text for each of the subheadings.

Tables 3 & 4

Text is difficult to read when centered. There needs to be adjustments to spacing to separate different headings . Table 4 contains this typo “incorrecSelf-Concept”

334 DISCUSSION

The first line of the Discussion should end with some references to support this statement.

Author Response

The Influence of Social Media on the Perception of Autism Spectrum Disorders: Content Analysis of Public Discourse on Youtube Videos

Thank you for the opportunity to review this interesting paper. My recommendation is to accept with minor changes. Here are some comments that need to be addressed.

REPLY  We thank you for your kind comment and thank you for having taken the time to review our manuscript.  We have addressed each issue that the reviewer has outlined and present an explanation below each.

This paper appears to adopt a medical model perspective on ASD. The language is replete with terms such disorder, symptoms, and even the term patient is used on line 52, with the term ‘treatment’ occurring 4 times. 

To make the paper more acceptable to a wider audience, they should take into consideration other perspectives on ASD. These references will be helpful

Leadbitter, K., Buckle, K. L., Ellis, C., & Dekker, M. (2021a). Autistic Self-Advocacy and the Neurodiversity Movement: Implications for Autism Early Intervention Research and Practice. Frontiers in Psychology12. https://doi.org/10.3389/fpsyg.2021.635690

Leadbitter, K., Buckle, K. L., Ellis, C., & Dekker, M. (2021b). Autistic Self-Advocacy and the Neurodiversity Movement: Implications for Autism Early Intervention Research and Practice. Frontiers in Psychology12. https://doi.org/10.3389/fpsyg.2021.635690

Milton, D. (2012). The normalisation agenda and the psycho-emotional disablement of autistic people. In Autonomy, the Critical Journal of Interdisciplinary Autism Studies (Vol. 1, Issue 1).

Silberman, S. (2015). Neurotribes: The legacy of autism and how to think smarter about people who think differently. Allen & Unwin.

REPLY

  • Thank you for suggesting that we take a larger perspective to make it more acceptable to a wider audience; we are also appreciative of the references provided. We agree and have made changes throughout the manuscript.  We have reduced the number of times we use ‘disorder’ and either altered the sentence or replaced the word to render it less based on the medical model.
  • We used the word “Treatment” in our first version as it encompasses, speech therapy, behaviour therapy etc. We have instead replaced it with “intervention”, as per Leadbitter et al 2021.
  • We have replaced terms such as disorder, symptoms, treatment, and patients with terms such as: cognitive, social and behavioural manifestations, intervention programs.

Lines 51-52

Regarding the term ‘patient’ on lines 51-52, it should be clarified that it is not the patient, but their behaviour that is used for making a diagnosis. This should be corrected.

REPLY

  • Thank you, we have made this change. We have changed the sentence to read “…behavioural observation”

111-112

It is implied here that ASD is a mental health condition, which it is not.

REPLY

  • Indeed, we agree. We have modified this instead to “…and neurodiversity” as guided by Leadbitter et al.

126

Tobin and Begley font needs resized.

REPLY

Thank you, this change has been made.

202

With respect to damaging language, the work by Kenny et al (2016) should be referenced here. This is important for the paper and for the analysis of terms. There may be cultural differences that need to be addressed.

“Recent public discussions suggest that there is much disagreement about the way autism is and should be described. This study sought to elicit the views and preferences of UK autism community members – autistic people, parents and their broader support network – about the terms they use to describe autism. In all, 3470 UK residents responded to an online survey on their preferred ways of describing autism and their rationale for such preferences. The results clearly show that people use many terms to describe autism. The most highly endorsed terms were ‘autism’ and ‘on the autism spectrum’, and to a lesser extent, ‘autism spectrum disorder’, for which there was consensus across community groups. The groups disagreed, however, on the use of several terms. The term ‘autistic’ was endorsed by a large percentage of autistic adults, family members/friends and parents but by considerably fewer professionals; ‘person with autism’ was endorsed by almost half of professionals but by fewer autistic adults and parents.”

Kenny et al. (2016). Which terms should be used to describe autism? Perspectives from the UK autism community. Autism, 20(4):442-62. DOI: 10.1177/1362361315588200

REPLY

  • Thank you for outlining this and providing the reference for Kenny et al. (2016)’s work; it has now been referenced. Indeed, there is no public consensus about the way autism is and should be described, especially that people use many terms to describe autism. (Kenny et al., 2016).

240

Text in left-hand column is difficult to read when centered.

REPLY

  • This has been modified. 

243-246

Use of capitalisation here is very odd.

REPLY

  • Fixed the capitalization issues.

Figures 1 & 2

The font is the pie chart could be reduced in size and this would allow the Key abbreviations to added to the chart. Not all Key terms are included in the figure caption.

REPLY

  • We have made the requested changes to the Figures, thank you.

256-257

“This 256 category can be further classified by the target audience of the video creator.”

REPLY

  • Explanation:
    The content creator’s “target audience” is the demographic of people with ASD the video focuses on. For example, the video is focused primarily on children with ASD or the video is focused on ASD in females.

    This, we have corrected the text to:

“….classified by the demographic of people with ASD the video focuses on:

Not clear what this means. Also, check the tense used in the text for each of the subheadings.

Tables 3 & 4

Text is difficult to read when centered. There needs to be adjustments to spacing to separate different headings . Table 4 contains this typo “incorrecSelf-Concept”

  • REPLY Modifications:

Left side of table no longer centered. Spaces added. Typos corrected

334 DISCUSSION

The first line of the Discussion should end with some references to support this statement.

  • REPLY Reference has been added.

Reviewer 3 Report

1. The authors must emphasize on the research gap and why those research questions are essential to be the focus on this article

2. The authors must provide more basis for determining YouTube as a powerful to raise awareness on ASD compared to other similar sites, since a lot users visit the site for entertainment purposes. Provide more sufficient explanation on the engagement level of current YouTube videos on raising awareness, is the available videos' on awareness has good response and have high visibility?

3. The authors collected the videos by not logging in, however the location of search affects the search results. Therefore the research results couldn't represent the global view of ASD, and the authors should give a limitation to the scope.

4. The authors should provide more details about determining the sentiment from videos, who determines the sentiment since it isn't done with a classifier like the text which uses a machine learning plugin.

Author Response

  1. The authors must emphasize on the research gap and why those research questions are essential to be the focus on this article
  • Thank you, we have tried to clarify this in the text by adding the following on Page 7:

There is a gap in the literature when it comes to the analysis of content together with the public’s reaction to ASD.  More evidence is needed to better inform how content with comments analysis on YouTube videos can improve the public’s perception of ASD. This will be achieved by answering the following research questions:

(1) what are the prevalent themes and sentiments in ASD related videos and their comments? (2) Do video presenters and commentators display stigmatization towards ASD?

  1. The authors must provide more basis for determining YouTube as a powerful to raise awareness on ASD compared to other similar sites, since a lot users visit the site for entertainment purposes. Provide more sufficient explanation on the engagement level of current YouTube videos on raising awareness, is the available videos' on awareness has good response and have high visibility?
  • Thank you, we have offered additional information on Page 6

“Notwithstanding the fact that the average number of views for the videos retrieved in 2019 were 826,156 and those retrieved in 2022 was 383,111 with some videos reaching millions of views, YouTube remains a go-to resource for viewing a variety of video types. Since its inception in 2005, it is one of the most visited sites in the world. Its content is diverse and global, and the platform offers the opportunity to disseminate content to a very broad audience (Khan, 2017). YouTube is an attractive platform for both amateur content creators and media companies, politicians, news organizations, businesses, music and film artists, and especially educational institutes (Khan, 2017), with many channels related to health – a quick search for ASD and YouTube shows over 33 million video results. “

  1. The authors collected the videos by not logging in, however the location of search affects the search results. Therefore the research results couldn't represent the global view of ASD, and the authors should give a limitation to the scope.
  • Indeed, by not logging in, it can limit search results, even though we used the rationale that signing into a personal account, may increase the influence of previous searches on the resulting videos. This important point has been added to the limitations section Page 19-20.
  1. The authors should provide more details about determining the sentiment from videos, who determines the sentiment since it isn't done with a classifier like the text which uses a machine learning plugin.
  • The mention of the Azure Machine Learning add-in for Excel is at the end of our methodological section right before the results section on Page 10. We have added to it to explain why this software is essential for this purpose.

  • “Microsoft Azure Machine learning was used to perform sentiment analysis and to develop a classification model that allows for the identification of sentiments (Harfoushi et al., 2018). It is now known that the Sentimental Analysis is a widely used technique in the natural language processing realm for determining the sentiment of a text, especially in social media. (Harfoushi et al., 2018).”